# Machine learning-based classification reveals distinct clusters of non-coding genomic allelic variations associated with Erm-mediated antibiotic resistance

Yongjun Tan,[1] Alexandre Le Scornet,[2] Mee-Ngan Frances Yap,[2] Dapeng Zhang[1,3]

**ABSTRACT** The erythromycin resistance RNA methyltransferase (*erm*) confers cross-resistance to all therapeutically important macrolides, lincosamides, and streptogramins (MLS phenotype). The expression of *erm* is often induced by the macrolide-mediated ribosome stalling in the upstream co-transcribed leader sequence, thereby triggering a conformational switch of the intergenic RNA hairpins to allow the translational initiation of *erm*. We investigated the evolutionary emergence of the upstream *erm* regulatory elements and the impact of allelic variation on erm expression and the MLS phenotype. Through systematic profiling of the upstream regulatory sequences across all known *erm* operons, we observed that specific *erm* subfamilies, such as *ermB* and *ermC*, have independently evolved distinct configurations of small upstream ORFs and palindromic repeats. A population-wide genomic analysis of the upstream *ermB* regions revealed substantial non-random allelic variation at numerous positions. Utilizing machine learning-based classification coupled with RNA structure modeling, we found that many alleles cooperatively influence the stability of alternative RNA hairpin structures formed by the palindromic repeats, which, in turn, affects the inducibility of *ermB* expression and MLS phenotypes. Subsequent experimental validation of 11 randomly selected variants demonstrated an impressive 91% accuracy in predicting MLS phenotypes. Furthermore, we uncovered a mixed distribution of MLS-sensitive and MLS-resistant *ermB* loci within the evolutionary tree, indicating repeated and independent evolution of MLS resistance. Taken together, this study not only elucidates the evolutionary processes driving the emergence and development of MLS resistance but also highlights the potential of using non-coding genomic allele data to predict antibiotic resistance phenotypes.

**IMPORTANCE** Antibiotic resistance (AR) poses a global health threat as the efficacy of available antibiotics has rapidly eroded due to the widespread transmission of AR genes. Using Erm-dependent MLS resistance as a model, this study highlights the significance of non-coding genomic allelic variations. Through a comprehensive analysis of upstream regulatory elements within the *erm* family, we elucidated the evolutionary emergence and development of AR mechanisms. Leveraging population-wide machine learning (ML)-based genomic analysis, we transformed substantial non-random allelic variations into discernible clusters of elements, enabling precise prediction of MLS phenotypes from non-coding regions. These findings offer deeper insight into AR evolution and demonstrate the potential of harnessing non-coding genomic allele data for accurately predicting AR phenotypes.

**KEYWORDS** antibiotic resistance, *ermB*, non-coding regulatory elements, MLS phenotype prediction, emergence and evolution of AR

Address correspondence to Mee-Ngan Frances Yap, frances.yap@northwestern.edu, or Dapeng Zhang, dapeng.zhang@slu.edu.

The authors declare no conflict of interest.

See the funding table on p. 20.

Antibiotic resistance (AR) is a global health threat, predicted to cause 10 million deaths and cumulative healthcare costs of $US100 trillion annually by 2050 if no new approaches are developed (1). In 2019, 4.95 million deaths were associated with bacterial AR worldwide, including 1.27 million deaths directly attributable to such resistance (2), highlighting the urgent need for improving surveillance, accurate AR forecast, and more effective prevention and therapies. Understanding the molecular mechanisms of AR (3) and systematically surveilling AR profiles in pathogens could lead to more accurate diagnostics and treatment, thereby mitigating the emergence of AR.

Much research has focused on identifying the presence of specific AR genes and their mutations within the coding regions (4–7), which have been the keys to predict AR phenotypes for bacterial strains with available genome data (5, 8–11). However, the impact of non-coding genomic allelic variations on the AR phenotypes is largely unexplored. Previously considered functionally inert, non-coding regions of the genome are now recognized as crucial regulatory elements, including promoter regions, untranslated regions, and intergenic regions. These elements influence factors such as transcription factor binding affinity, RNA stability, and ribosome and antisense RNA accessibility (12–16). Consequently, allelic variations in these regions can lead to alterations in the expression patterns of AR genes, thereby affecting antibiotic susceptibility (17). A deeper understanding of the relationship between non-coding genomic allelic variations and AR phenotypes will enhance our ability to accurately predict bacterial resistance and enable more precise and tailored approaches to antibiotic treatments.

One of the most compelling examples of non-coding region regulation in AR mechanisms comes from the study of the erythromycin-resistance methylase (*erm*) family of RNA methyltransferases (18, 19). These enzymes catalyze the dimethylation of the conserved nucleotide A2058 within the 23S rRNA during ribosome maturation (19). While this modification, known as m$^6$A2058, has negative effects on general translation and host colonization in bacterial cells (20), it imparts resistance to antibiotics belonging to the macrolides, lincosamides, and streptogramins (MLS) classes (21, 22). MLS antibiotics are structurally distinct translation inhibitors that primarily target the 50S large subunit of the bacterial ribosome. When exposed to a sublethal concentration of macrolide, bacteria activate the expression of *erm* genes to install m$^A$2058 to the ribosomes. The m$^6$A group sterically hinders the binding of MLS antibiotics to the ribosome (23), consequently leading to cross-resistance against all MLS antibiotics. Studies on various *erm* genes (19, 24–27), such as *ermA*, *B*, *C*, *D*, and others, suggest that macrolide-inducible expression of *erm* is primarily regulated by their upstream regulatory region through a mechanism known as "translational attenuation" or "ribosome stalling" (28, 29). These 5′ regulatory elements typically feature a short upstream ribosome-stalling leader peptide open reading frame (uORF; e.g., *ermBL* in the *ermBL-ermB* operon), and potential RNA structures that regulate the accessibility of ribosome to the start codon of the downstream *erm* gene.

A possible model can be learned from the case of *ermB* regulation (30), which involves two distinct regulatory stages (Fig. 1). Under conditions without antibiotics (referred to as the "repression stage"), an RNA secondary structure termed "hairpin-2" forms near the translation start codon of the *ermB* open reading frame (ORF). This hairpin-2 structure impedes translation by blocking ribosome access to the start codon, resulting in basal *ermB* gene expression. By contrast, in the presence of macrolide antibiotics, ribosomes binding to these antibiotics experience functional disruption and become arrested at the upstream ErmBL leader sequence of the *ermBL-ermB* transcript. The translation arrest facilitates the formation of a secondary structure known as "hairpin-1". Importantly, the formation of hairpin-1 prevents the subsequent formation of the downstream hairpin-2 structure, as they share a critical RNA sequence element required for their stability. As a consequence, the translation start codon of *ermB* becomes exposed to the ribosome, facilitating translational initiation of *ermB* and conferring antibiotic resistance.

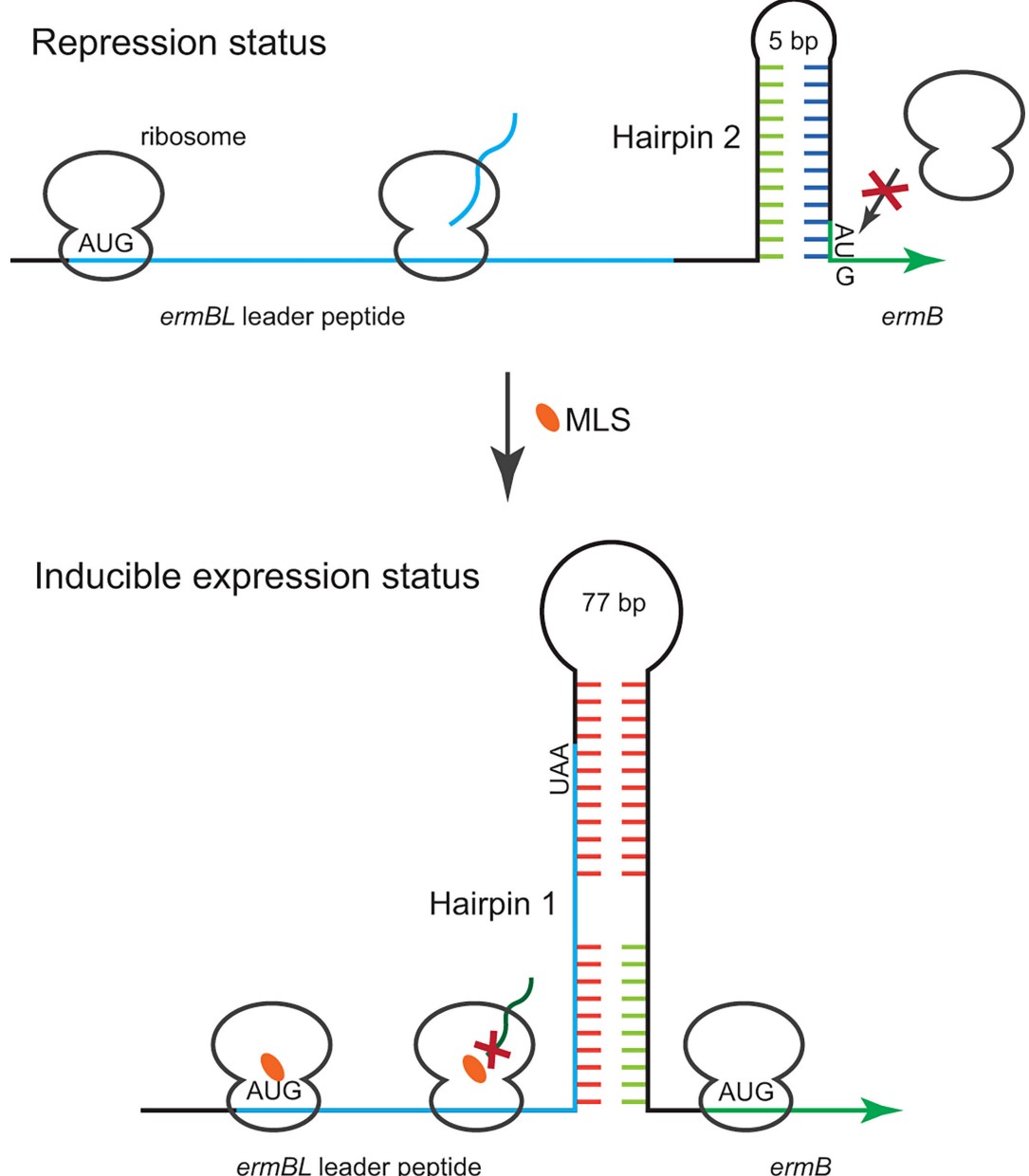

**FIG 1** Schematic for the ribosome stalling mechanism for *ermB* expression during both the repression stage (top) and the inducible expression stage (bottom). The schematic is drawn based on the reference *ermBL-ermB* region (EF450709.1: 2096-3041). The small upstream ORF (uORF) encoding the *ermBL* leader peptide is highlighted in cyan, the *ermB* coding region is depicted in green, nucleotides forming hairpin-1 are marked in red, nucleotides shared by both hairpin-1 and hairpin-2 are shown in light green, and the remaining nucleotides in hairpin-2 are represented in dark blue.

This process is thus referred to as the "antibiotics-inducible expression stage". Similar mechanisms may apply to other *erm* genes, although experimental validation is needed.

Beyond the major mechanism that involves the dynamic interactions between ribosome, antibiotics, and the 5′ regulatory region, mutagenesis experiments suggest that mutations on the 5′ regulatory region can regulate *erm* expression and influence the AR phenotype. Focus has been put on the various nucleotide positions on the short uORFs and their effect on translational activation during antibiotic induction (25–27, 29–31). However, a systematic understanding of allelic variation, especially those naturally occurring ones, in the *erm* expression regulation is still lacking. In our study, we have conducted a comprehensive investigation into all *erm* genes to understand the

link between non-coding regulatory regions and AR phenotypes. Specifically, we aim to address several key questions regarding the MLS antibiotic resistance emergence and evolution: (1) how did the MLS-inducible mechanism emerge during evolution? (2) how does allelic variation in the 5′ non-coding regulatory elements affect AR phenotypes? (3) what role do evolution and selection play in the AR mechanism? and (4) can MLS-AR phenotypes be predicted from non-coding genomic data? This study is significant because *erm* genes are among the high-risk AR gene families classified by the World Health Organization (WHO) based on their abundance, mobility, and association with bacterial pathogenicity (32–35)

## MATERIALS AND METHODS

### Protein sequence search, alignment, and phylogenetic analysis

To collect *erm* protein homologs, iterative sequence profile searches were conducted using the Position-Specific Iterated BLAST (PSI-BLAST) program (version 2.13.0+) (36) against the non-redundant (nr) protein database of NCBI with a cut-off e-value of 0.005 serving as the significance threshold. Similarity-based clustering was performed by BLASTCLUST, a BLAST score-based single-linkage clustering method (ftp.ncbi.nih.gov/blast/documents/blastclust.html). Protein multiple sequence alignments (MSA) were built by the PROMALS3D program (37), followed by careful manual adjustments. Phylogenetic analysis was conducted using the FastTree (38) and MEGA7 (39) programs.

### Detection of sequence features in the *erm* upstream regions

To identify the small uORFs and upstream palindromic repeat regions associated with *erm* genes and homologs, the following procedure was employed: we first extracted the upstream nucleotide sequences of all known *erm* genes. This involved retrieving the genomic loci information for *erm* genes and their homologous based on the *erm* phylogeny. Specifically, nucleotide sequences encompassing the *erm* gene upstream region (~250 bp) and the first 10 positions of the *erm* coding region were extracted. Then the extracted upstream sequences were analyzed using the ORFfinder program (https://www.ncbi.nlm.nih.gov/orffinder/) to identify potential open reading frames (ORFs). Only the ORFs located on the same strand as their corresponding *erm* genes were considered, as they are co-transcribed with the corresponding *erm* genes. Next, to detect the potential upstream palindromic repeat regions, the BLASTN program (version 2.13.0+) (40) was employed to conduct pairwise comparisons for each *erm* upstream sequence. The comparison utilized a word size of 7 and an e-value threshold of 10. Palindromic repeat regions were identified as inverted repeats located on different strands, with one repeat on the forward strand and the other on the reverse strand, as revealed by the BLASTN results.

### Identification of population-wide allelic variation of upstream regions of *ermB*

To study the function and natural occurrence of different alleles combinations in *ermB* upstream region among bacterial populations, we utilized population-wide genome analysis (Fig. 3A).

First, to make a comprehensive collection of *ermB* upstream nucleotide sequences, we used two sequence search strategies *via* BLASTP (version 2.13.0+) and BLASTN programs (version 2.13.0+) (40). At first, we collected *ermB* protein homologs by using *ermB* sequence (ABQ00061.1) as query searching against NCBI non-redundant protein (nr) database *via* the BLASTP program with a cut-off e-value of 0.005 serving as the significance threshold and retrieved 6,684 significant hits. Next, we extracted their genomic loci information from the NCBI GenBank database using the NCBI EDirect programs (https://www.ncbi.nlm.nih.gov/books/NBK179288/) and retrieved the

nucleotide sequences containing the *erm* gene upstream region (~250 bp) and the complete *ermB* coding region.

Second, we used the *ermB* gene region (covering the *ermBL-ermB* operon nucleotide sequence; GenBank EF450709.1:2096-3041 nt) to search against either the NCBI non-redundant nucleotide (nt) database or the NCBI whole-genomes shotgun contigs (WGS) database using BLASTN with a word size of 11 and e-value 0.001 severing as significant cut-off and retrieved the aligned subject sequences. After removing the sequences from the same genomic loci, we merged the above nucleotide sequences and built an *ermB* upstream nucleotide sequence BLAST database.

Subsequently, to remove any potential non-*ermB* sequences, we conducted another search using the BLASTN program against the previously built database. This time, we exclusively retained sequences that exhibited alignment hits with the upstream region of *ermB*, specifically, GenBank EF450709.1:2096-3041 nt. Following this filtering step, we proceeded to create a multiple sequence alignment (MSA) comprising a total of 21,525 sequences based on the BLASTN result (output format 1) and calculated the frequency of mutations, deletions, and insertions for *ermB* upstream region using a custom Python3 script (see Data Availability). With that, we draw the stacked column graph to show the variation frequency percentage in each position of the *ermB* gene upstream region (Fig. 3B).

To gather the animal host information for the bacterial genomes containing the identified *ermB* upstream element variants, we employed the NCBI EDirect programs (https://www.ncbi.nlm.nih.gov/books/NBK179288/) to retrieve GenBank files and extract relevant host keywords.

## Entropy analysis

Position-wise Shannon entropy (*H*) for a given nucleotide multiple sequence alignment was calculated using the equation:

$$H = -\sum_{i=1}^{M} Pi \log_2 Pi$$

*P* is the probability of each nucleotide *i*, and *M* is the number of nucleotide types. The Shannon entropy for the ith position in the alignment ranges from 0 (only one nucleotide at that position) to 2 (all four nucleotides equally represented at that position) in a four-letter alphabet. Analysis of the entropy values which were thus derived was performed using Python3.

## Machine learning-based classification

To acquire the natural occurrence of different allele combinations, we made a classification for all sequences upstream of *ermB* gene (1–211 bp). Due to high sequence similarity, we could not employ traditional clustering methods to classify them into different clusters. Thus, we conducted a clustering strategy with machine learning algorithms (Fig. 4A). Based on the *ermB* upstream sequence alignment, we first transformed the alignment data using the One-Hot encoding method to make it suitable for machine learning algorithms. In the alignment, nucleotide types are categorical data represented by A, T, G, C, and gap (-). One Hot encoding converts each category into a distinct numerical vector: A is encoded as (1,0,0,0); T as (0,1,0,0); C as (0,0,1,0); G as (0,0,0,1); and Gap (-) as (0,0,0,0). We then conducted dimensionality reduction using principal component analysis (PCA) approach with 0.95 as n_components value to keep 95% alleles and accelerate calculating speed.

Next, we used a distance-based unsupervised clustering algorithm, K-means algorithm, to make a clustering. To determine the optimal number of clusters for the K-means algorithm, we conducted a comprehensive series of clustering trials involving cluster numbers ranging from 20 to 100. Each trial was assessed through the computation of two key metrics: the average (mean) of alleles (mutations) across clusters and

the average (mean) of normalized alleles (mutations) across clusters. The average (mean) of alleles across clusters was computed by aggregating the alleles in all clusters and subsequently dividing this sum by the total number of the clusters. Conversely, the average (mean) of normalized alleles across clusters was calculated by summing the ratios of number of the alleles to the number of the sequences within each cluster, followed by dividing the resulting sum by the total number of clusters. The classification program is available on GitHub (see Data Availability).

Utilizing these two criteria, we determined the optimal cluster number to be 59 (Fig. 4B). Subsequently, employing this optimal cluster number, we performed a cluster-ing analysis utilizing the K-means algorithm. This analysis allowed us to identify the prevalent allele combinations within each cluster, along with their respective frequen-cies. In addition, we constructed a Multiple Sequence Alignment (MSA) for representative sequences from each cluster using Kalign (version 3.3.2) (41). Subsequently, the MSA was color-coded using the CHROMA program (42) and underwent further refinement using Adobe Illustrator.

## RNA secondary structure prediction and analysis

To examine the potential effect of different alleles on the antibiotic-inducible *ermB* expression, we predicted the RNA secondary structure for the representative sequence from every cluster using the RNAfold program of the ViennaRNA Package (version 2.6.3) (43) and evaluated their effect on the formation of the alternative hairpin structures based on the free energy, which was calculated using the Andronescu model (44). Specifically, we used full-length *ermB* upstream sequences (1–211 bp) to predict the RNA secondary structure, resulting in all sequences only forming hairpin-2 structure but not hairpin-1. Thus, we next used the truncated version, which deleted the last 12 nucleotides in *ermB* upstream sequences (1–199 bp), to prevent the formation of the hairpin-2 and allow prediction for the hairpin-1 structure. Finally, the RNA secondary structure of *ermB* upstream sequences, hairpin-1 and -2 structures, was colored and further modified using Adobe Illustrator.

## Construction of upstream *ermB* mutant variant strains and growth conditions

Strain JE2 is an *erm*-deficient community-associated methicillin-resistant *Staphylococcus aureus* (CA-MRSA) of USA300 lineage (45). To construct *ermBL-ermB* proficient strains, the *ermBL-ermB* region, including its native promoter, was PCR amplified from strain CM05 (GenBank EF450709) with primers P1186_PstI (5′- ATCTGCAGTTGGTCTTGCGTATGGTTAA CCCTAAAG-3′) and P1187_KpnI (5′-TAGGTACCTAGAATTATTTCCTCCCGTTAAATAATAGA-3′) and cloned into the PstI and KpnI sites of the CdCl$_2$-resistant suicide vector, pJC1111 (46). The suicide plasmid was integrated into the chromosome of strain RN9011 following standard protocols (46). The *ermBL-ermB* gene region was subsequently transferred to the JE2 strain *via* φ11 phage transductions, resulting in KES29 bearing the single-copy *ermBL-ermB*. To construct KES30d, A site-directed Quikchange mutagenesis kit (Agilent Genomics) was used to introduce *ermBL*(R7stop) into the pJC1111::*ermBL-ermB* gateway plasmid. To construct variants 1–11 (Table 1), synthetic DNA fragments of different *ermBL-ermB* variants were purchased from Twist BioScience and cloned into pJC1111 as described above. Unless otherwise noted, *S. aureus* cells were grown aerobically at 37°C in tryptic soy broth (TSB, BD Difco #211822) at a 5:1–10:1 tube- or flask-to-medium ratio with a 1:100 dilution of an overnight seed culture. All chemicals were from Sigma-Aldrich unless otherwise noted. CdCl$_2$ was used at the final 0.15 mM. Primers were purchased from IDT DNA.

## Measurement of minimum inhibitory concentration

Minimum inhibitory concentration (MIC) values of erythromycin and clindamycin were determined by E-test strips (Biomerieux) on the Mueller Hinton agar (BD Difco #225250) plates following the manufacturer's manual. MICs were recorded after 24-h incubation at 37°C.

**TABLE 1** The impact of ermB upstream sequence polymorphism on macrolide (erythromycin, ERY) and lincosamide (clindamycin, CLN) resistance[j]

| ermBL–ermB[a] | Cluster[b] | Frequency[c] | nt change[d] | a.a. change[e] | Free energy of Hairpin-1[f] | Free energy of Hairpin-2[f] | ErmB expression inducibility[g] | ERY[h] | CLN[h] | Antibiotic resistance[i] (experiment) |
|---|---|---|---|---|---|---|---|---|---|---|
| None (JE2) | — | — | — | — | — | — | — | 0.19 | 0.032 | Extremely sensitive |
| WT (KES29) | Cluster_24 | 103 | — | — | −17.69 | −12.15 | Reference | 12 | 1.5 | Sensitive (control) |
| KES30d | | | C19T, G20G, U21A | R7stop | | | — | >256 | >256 | Resistant (control) |
| V1 | Cluster_11 | 222 | C13A, T22A, G176A, **U181C** | Q5K | −16.65 ↓ | −12.15 – | Weak | 12 | 1.5 | Sensitive |
| V2 | Cluster_18 | 174 | T22A, G176A, **U181C**, *U202A* | Y7N | −16.65 ↓ | −7.4 ↓ | High | >256 | >256 | Resistant |
| V3 | Cluster_37 | 186 | U22A, G176A, **U181C**, *C191A* | Y7N | −11.48 ↓ | −6.98 ↓ | High | >256 | >256 | Resistant |
| V4 | Cluster_0 | 2,914 | U22A, G176A, **U181C** | Y7N | −16.65 ↓ | −12.15 – | Weak | 12 | 1.5 | Sensitive |
| V5 | Cluster_8 | 623 | U22A, G73A, G176A, **U181C** | Y7N, A25T | −17.58 ↓ | −12.15 – | Weak | >256 | >256 | Resistant |
| V6 | Cluster_13 | 304 | U22A, G73A, G176A | Y7N, A25T | −19.64 ↑ | −12.15 – | High | >256 | >256 | Resistant |
| V7 | Cluster_53 | 4 | C13A, U22A, G73A, *U206A* | Q5K, Y7N, A25T | −18.62 ↑ | −9.27 ↑ | High | >256 | >256 | Resistant |
| V8 | Cluster_2–2 | 143 | U22A, *C74T*, G176A | Y7N, A25V | −19.45 ↑ | −12.15 – | High | >256 | >256 | Resistant |
| V9 | Cluster_2–1 | 3,781 | U22A, G176A | Y7N | −18.71 ↑ | −12.15 – | High | >256 | >256 | Resistant |
| V10 | Cluster_21 | 54 | U22A, G176A, **U181C**, *G203A* | Y7N | −16.65 ↓ | −6.98 ↓ | High | >256 | >256 | Resistant |
| V11 | Cluster_1 | 5,289 | Δ1–12, C63A, *G203A*, *Insert63A* | Δ(MLVF), N21K | −18.80 ↑ | −6.10 ↓ | High | >256 | >256 | Resistant |

[a]S. aureus JE2 carrying the chromosomal ermBL–ermB operon (GenBank: EF450709) with mutated upstream elements.
[b]The representative sequences from randomly selected clusters are validated using experimental methods.
[c]The frequency of genomes that contain each type of the upstream sequence clusters.
[d]Nucleotide (nt) substitutions in the ermB upstream region (ermBL–ermB). Nucleotide numbering follows the ermBL open reading frame with the A in the start codon ATG designated as position 1. The substitution that destabilizes mRNA hairpin 1 and hairpin 2 is indicated in bold font and italic font, respectively. The substitution that stabilizes the hairpin 1 is marked with an underline.
[e]Amino acid changes in the ErmBL due to non-synonymous mutations.
[f]The free energy of hairpin-1 or hairpin-2 structures was calculated by the RNAfold program (43). Increased stability is indicated by the up arrow and decreased stability by the down arrow.
[g]The predicted ermB expression inducibility compared to the reference cluster, WT(KES29).
[h]The minimum inhibitory concentrations of antibiotics (ERY, CLN) on S. aureus cells transferred with various ermB upstream sequence variants.
[i]Experimentally determined antibiotic resistance phenotypes.
[j]Minimum inhibitory concentrations (MIC, μg/mL) were determined by E-test on Muller Hinton agar plates in biological triplicates per strain per antibiotic.

## Induction of *ermB* expression by erythromycin and western blots

*S. aureus* cells were treated with a sublethal concentration of erythromycin (1 µg/mL), or 70% ethanol mock when density reached $OD_{600}$ ~ 0.6. Erythromycin induction of *ermB* expression was continued at 37°C for 90 min before harvesting. Cell pellets were homogenized with Lysing Matrix B (MP Biomedicals) in 25 mM Tris (pH 7.5) on a FastPrep-24 homogenizer (MP Biomedicals). Clarified lysates were recovered by spinning at 20,817 × *g* at room temperature for 5 min to remove cell debris. A total of 0.1–0.2 $Abs_{280}$ units of cell lysate were analyzed on 4%–20% TGX SDS-PAGE gels (BioRad) and the proteins were transferred to a nitrocellulose membrane using a Trans-Blot Turbo system (BioRad). The membrane was stained with Ponceau red (Amresco #K793-500mL) to ensure equal loading, followed by immunoblotting using 1/1,000 dilution of anti-*ermB* generated using *Clostridium perfringens ermB* as the antigen (kindly provided by J. Rood) (30, 47). HRP-conjugated anti-IgG secondary antibody (1/15,000 dilutions) was from Cytiva (#NA9120, 1/15,000 dilutions). SuperSignal West Dura chemiluminescence substrate was used (Thermo Scientific #34075). Images were acquired using iBright FL1500 system (ThermoFisher).

## RESULTS

### Evolutionary history of the *erm* family and the independent emergence of uORFs and palindromic repeats

Previous studies have shown that two sequence features, namely the uORFs and the palindromic repeats, in the 5′ upstream regions of *erm* genes are essential for antibiotic-inducible regulation (29). To investigate these features in an evolutionary context, we initiated a comprehensive analysis of 23S rRNA methyltransferases, which included 35 known *erm* genes (48) and their homologs retrieved using PSI-BLAST searches against the NCBI nr database. Given that all known *erm* genes belong to the 23S rRNA methyltransferase family, we also gathered the related 16S rRNA adenine methylases to serve as an outgroup for *erm* genes to establish the evolutionary tree. Upon constructing the evolutionary tree, it became apparent that all 23S rRNA methyltransferases shared a distinct relationship with 16S rRNA adenine methylases. However, the experimentally verified *erm* genes did not form a monophyletic clade (Fig. 2A). This observation suggests that other 23S rRNA methyltransferase genes share the same function as *erm* antibiotic resistance genes.

With the phylogeny in mind, we proceeded to investigate the presence of uORFs among the upstream regions (~250 bp) of *erm* genes. Out of the 35 known *erm* genes, only 14 had been annotated to include uORFs in their upstream regions by the Refseq database (49). These included examples such as *ermA*, *ermB*, *ermC*, *ermD*, and others. To identify potential uORFs in other *erm* loci, we extracted the upstream nucleotide sequences of the remaining 43 *erm* homologs and predicted the presence of upstream small ORFs using the ORFfinder program (https://www.ncbi.nlm.nih.gov/orffinder/). This analysis revealed an additional 21 *erm* homologs containing small ORFs in their upstream regions (Fig. 1B; Table S1). Importantly, not all *erm* genes possessed uORFs, indicating that the presence of small uORFs in gene regulation is not a universal trait among *erm* genes. In addition, we performed pair-wise sequence comparisons at both nucleotide and protein levels for the identified uORFs, revealing high divergence. Some of these uORFs shared sequence similarity and formed individual families, but no universal conservation pattern could be identified across all families (Fig. 2A and B). This suggests that uORFs are not universally conserved throughout evolution.

Subsequently, we explored the presence of palindromic repeat patterns in the upstream regions of all *erm* genes using the BLASTN program. Only 21 *erm* genes were found to contain an upstream palindromic region (Fig. 2A), and merely 18 *erm* genes exhibited both uORFs and palindromic regions. This observation indicated that similar to uORFs, the presence of palindromic regions is not a universal feature in the upstream regions of all *erm* genes. Notably, the 18 *erm* genes harboring both uORFs

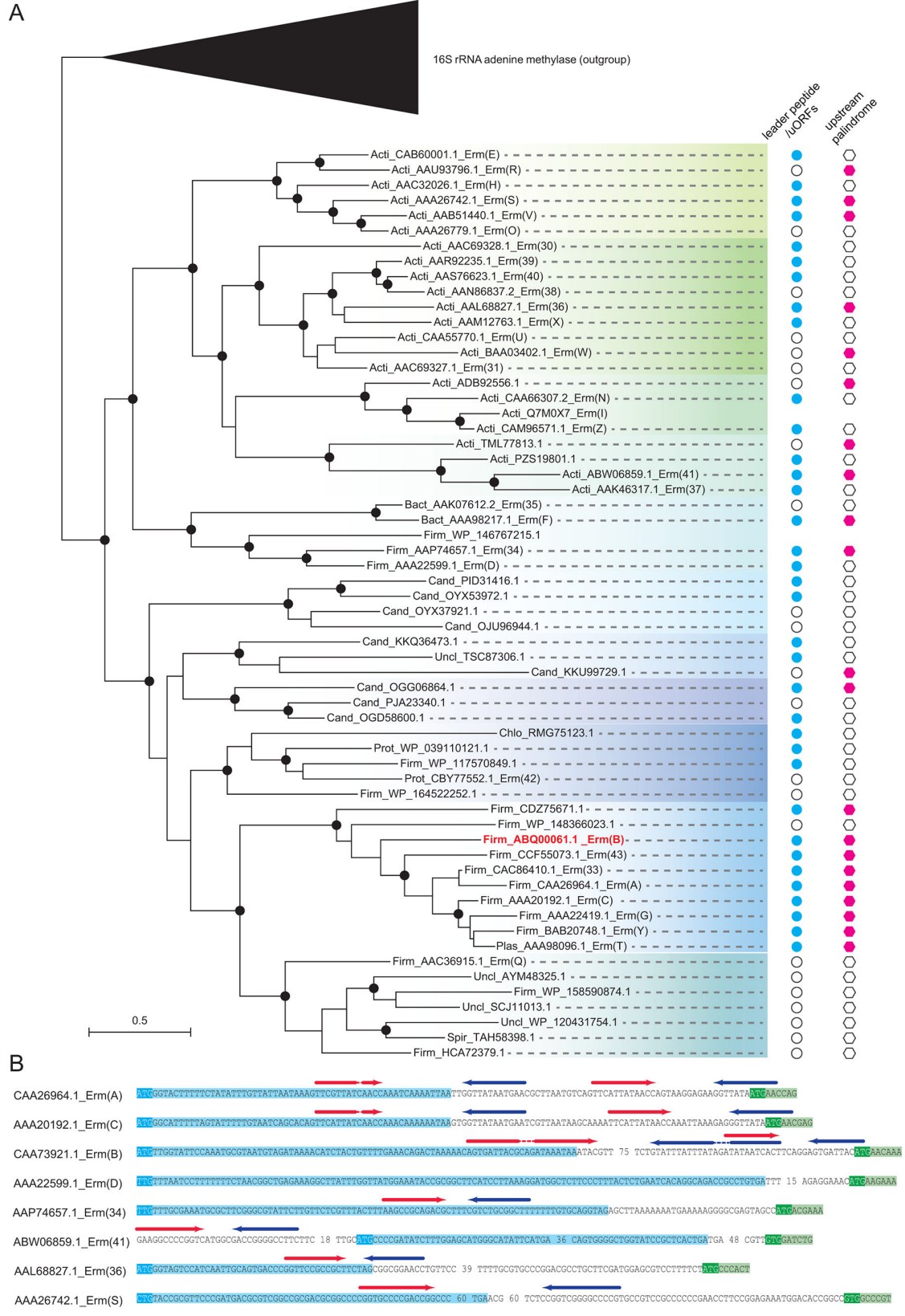

FIG 2 Phylogenetic analysis of the *erm* family and distribution of uORFs and upstream palindromic regions. (A) The phylogeny was inferred by the FastTree program. The tree with the highest log-likelihood is presented, in which the internal nodes with significantly supported values are labeled in dark dots. The scale bar represents 0.5 substitutions per site. Each sequence is labeled with its phylum followed by NCBI protein accession number. The previously defined

**FIG 2** (Continued)

*erm* genes are also highlighted with their corresponding gene name following the NCBI accession numbers. The distribution of leader peptides/uORFs and upstream palindromic regions of corresponding *erm* sequences are displayed on the right side of *erm* phylogeny and highlighted in sky blue dots and magenta hexagons, respectively. Open circles and hexagons represent the absence of either the leader peptides/uORFs or the upstream palindromic regions. Species name abbreviations: Acti, Actinomycetes; Bact, Bacteroides; Firm, Firmicutes; Cand, Candidatus; Chlo, Chloroflexota; Prot, Pasteurella; Plas, Lactobacillus; Uncl, Unclassified sp; Spir, Sphaerochaeta. (B) Representative examples of *erm* genes that have distinct sets of uORFs and/or palindromic repeats. In each pair of hairpin-forming repeats, the forward sequence is highlighted in red and the reverse sequence in blue, with dotted lines indicating the unpaired regions.

and upstream palindromic regions were scattered throughout the phylogenetic tree (Fig. 2A). Furthermore, nucleotide sequence analysis revealed significant divergence among palindromic sequences, with some showing high similarity and others being highly divergent. This highlights that upstream palindromic regions are not universally conserved over extended evolutionary timescales.

Based on the distribution of uORFs and palindromic regions, we postulate that these elements within the upstream regions of *erm* may have evolved independently. Notably, experimental evidence has demonstrated that mutations removing the uORF result in the constitutive expression of *erm* genes. In light of our recent discovery that *erm* has a detrimental impact on bacterial translation (20), it is plausible that the inducible expression of *erm* genes represents an outcome of selection pressure (50). This selection may have harnessed the independent evolution of uORFs and palindromic repeats to counteract the negative effects on bacterial fitness while concurrently achieving antibiotic resistance as a response to antibiotic pressure.

## Population-wide genomic allele frequency analysis of the *ermB* upstream region

Given the independent emergence of upstream sequence elements in different *erm* genes, we focused our investigation on the *ermB* subfamily. The 5′ upstream region of *ermB* genes features a conserved configuration of a short uORF encoding ErmBL and two palindromic repeat regions. Therefore, we directed our analysis towards the region spanning from the start codon of the uORF to the start codon of the *ermB* gene (corresponding to the *ermBL-ermB* operon nucleotide sequence in the reference locus EF450709.1:2096-3041). Our goal was to uncover the hallmarks of evolutionary selection operating on these upstream regions and to gain insights into their functional roles. To accomplish this, we employed two strategies (Fig. 3A) to assemble an extensive collection of *ermB* upstream sequences, totaling 21,525 sequences that encode *ermB* genes and share significant similarities with the upstream regions of the above reference *ermB*. Subsequently, we constructed a multiple sequence alignment (MSA) for these sequences and calculated the allele frequencies, encompassing mutations, deletions, and insertions, for each position within the *ermB* upstream regions (Fig. 3B). The analysis revealed that most positions within the *ermB* upstream elements remain highly conserved. However, several positions display elevated variation (hyper alleles), particularly within the uORF and palindromic regions, including those corresponding to reference C13, U22, U42, C63, G73, G159, G176, U181, C191, U202, G203, U206, C208, and one insertion between positions 202 and 203 (Fig. 3B). Furthermore, our examination included Shannon entropy analysis, which indicated that these variable positions display a substantial degree of variability (complexity) with entropy values ranging between 0.1 and 1.0, as opposed to a random distribution (average entropy of the region: 0.04; medium entropy of the region: 0.0039) (Fig. 3C). This observation supports the notion that these positions have been subject to significant functional selection for diversity, implying their crucial role in the regulatory mechanism.

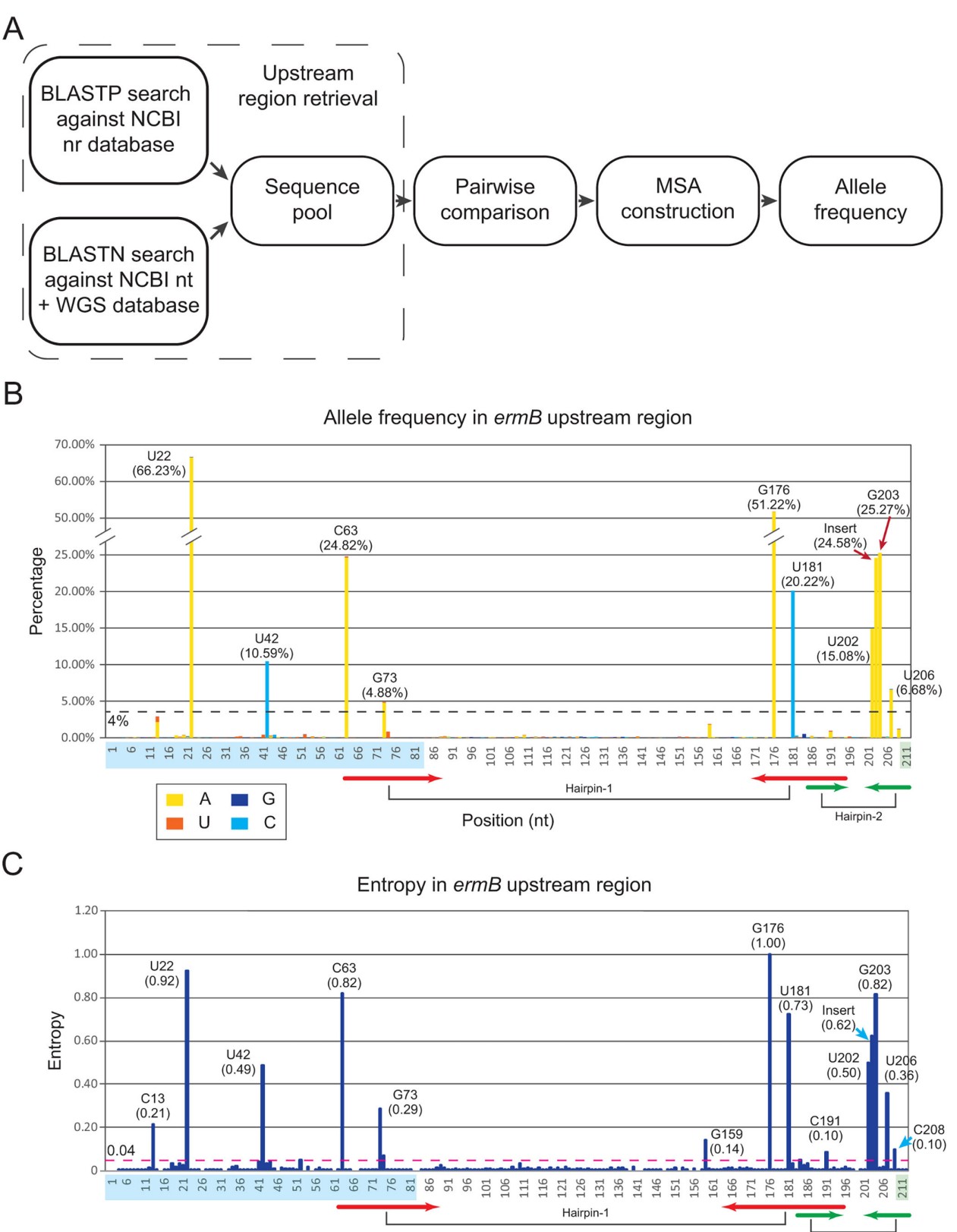

**FIG 3** Allelic variations in the *ermB* upstream region. (A) The conceptual diagram illustrating the steps of population-wide allele frequency analysis used in this study. (B) Stacked column graph of the percentage of allele frequency of *ermB* upstream region. Compared with the reference sequence (EF450709.1: 2096-2306), each column represents the frequency of the variations in each position. The frequency percentages of adenine (A) are shown in yellow, thymine (T) in orange,

**FIG 3** (Continued)

guanine (G) in dark blue, and cytosine (C) in cyan. The reference nucleotide type and the percentage of the non-reference variations for the dominant positions are labeled at the top of each column. The *ermBL* uORF is highlighted in a light blue background on the X-axis, and the start codon of *ermB* is colored in light green. Forward and reverse sequences in the hairpin-1 structure are present in red arrows, while the forward and reverse sequences in the hairpin-2 structure are in green arrows. (C) Entropy plot for the *ermB* upstream region. Shannon entropy data were computed for each position for a character space of four nucleotides. The mean entropy across the entire length of the upstream region is indicated as a dashed red horizontal line. Where a position shows high entropy, it is a sign of potential selection for diversification.

## Clustering analysis of the *ermB* upstream regions using the machine learning algorithm

As most hyperalleles are located in the palindromic regions, we argue that these alleles might influence the formation of the RNA structures (hairpins) which will further modulate *ermB* gene regulation. However, the large number of such alleles prevented us from conducting the mutagenesis experiment extensively. Furthermore, there is a possibility that different alleles might have an interaction. Therefore, we sought to conduct a clustering analysis of these sequences, aiming to identify the major groups of sequences that share high sequence similarity and allele combinations. Due to high sequence similarity, traditional clustering methods cannot classify the *ermB* upstream sequences effectively. To tackle this problem, we developed a machine learning approach, in which nucleotide MSA will be used as input into three sequential steps including data decoding, PCA dimension deduction, and k-means clustering (Fig. 4A, and details in methods). By minimizing the average (mean) of alleles (mutations) across clusters and the average (mean) of normalized alleles (mutations) across clusters (details in methods), we have successfully converted 21,525 sequences into 59 distinct sequence clusters (Fig. 4B).

Figure 4C depicts a multiple sequence alignment of the representative sequences from each cluster, accompanied by the associated frequency. The alignment not only illuminates the prevailing combinations of various alleles within distinct clusters but also unveils sporadic mutations in the upstream region of *ermB* within the clusters. Furthermore, many hyper alleles identified above are also shared across different clusters, each manifesting a unique combination of these alleles. In addition, it is noteworthy that a majority of the clusters contain the alleles within the palindromic repeat regions, suggesting a potential influence of these alleles on the formation of hairpin structures in the *ermBL-ermB* transcript.

## Various alleles regulate *ermB* gene expression by modulating the formation and stability of alternative hairpin structures

We next sought to investigate the impact of various alleles in palindromic repeat regions on hairpin structure formation and its subsequent effect on *ermB* gene regulation. To test this, we used RNA secondary structure prediction to analyze hairpin structure formation and stability for all identified sequence clusters. Based on the antibiotic-inducible ribosome stalling model, the differential regulation of *ermB* expression is associated with the formation of the alternative hairpins, hairpin-1 or hairpin-2. Under conditions without antibiotics (repression stage), hairpin-2 is formed, blocking the ribosome's access to the *ermB* ORF translation start codon, resulting in no *ermB* expression. Conversely, in the presence of antibiotics (inducible expression stage), hairpin-1 formation prevents hairpin-2 formation, leading to *ermB* expression and consequent MLS resistance. Thus, we focused on examining the potential of identified clusters, containing various alleles, to form these hairpin structures by calculating their local free energy. This energy indicates RNA secondary structure stability, with higher values signifying lower stability.

Our findings showed that most clusters can form both hairpin structures, yet they varied in formation potential and stability (Table S2; Fig. 5). For hairpin-1 structures, while most clusters exhibited similar free energy levels compared to the reference variant (cluster_24), others with various allele combinations display either increased

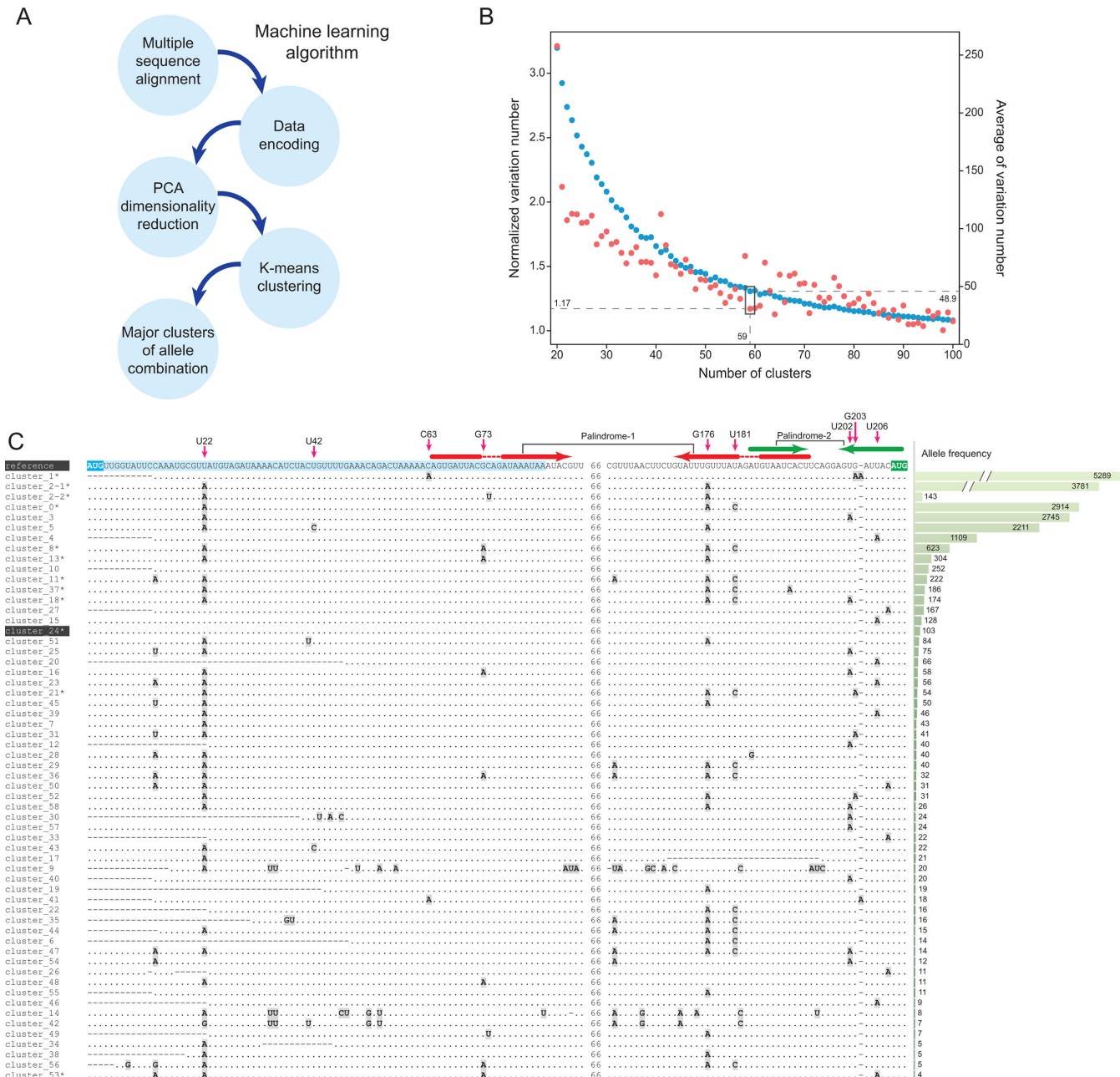

**FIG 4** Clustering analysis of the *ermB* upstream sequences. (A) The conceptual diagram illustrates the major steps of clustering analysis of *ermB* upstream sequences using the machine learning algorithm. (B) Dot plot shows the average of normalized variation number (red) and the average of variation number (blue) among different cluster numbers. The optimal cluster number used in this study is 59. (C) Multiple sequence alignment of representative sequences from major clusters of *ermB* upstream region. The reference sequence (EF450709.1: 2096-2306 nt) is shown at the top of the alignment followed by the representative sequences from major clusters. Note that the reference sequence is part of Cluster_24, as indicated by the grey shading. The coding region of *ermB* leader peptide is highlighted in a light blue background in the reference sequence, and the start codons of *ermB* leader peptide and *ermB* gene are colored in cyan and green, respectively. The representative sequences from major clusters are labeled with their corresponding cluster number, in which the sequences with an asterisk (*) are validated using experimental methods. In the alignment, each dot (.) indicates that this position is identical to the corresponding position in the reference sequence, while A, U, G, and C with gray backgrounds represent the variations found in each cluster compared with the reference sequence. The dashed line (-) indicates the gap in the alignment. The reference positions containing dominant variations and the palindromic region are highlighted at the top of the alignment. Forward and reverse sequences in the hairpin-1 structure are shown in red arrows, while the forward and reverse sequences in the hairpin-2 structure are in green arrows. The frequency of each cluster is displayed on the right panel of the alignment.

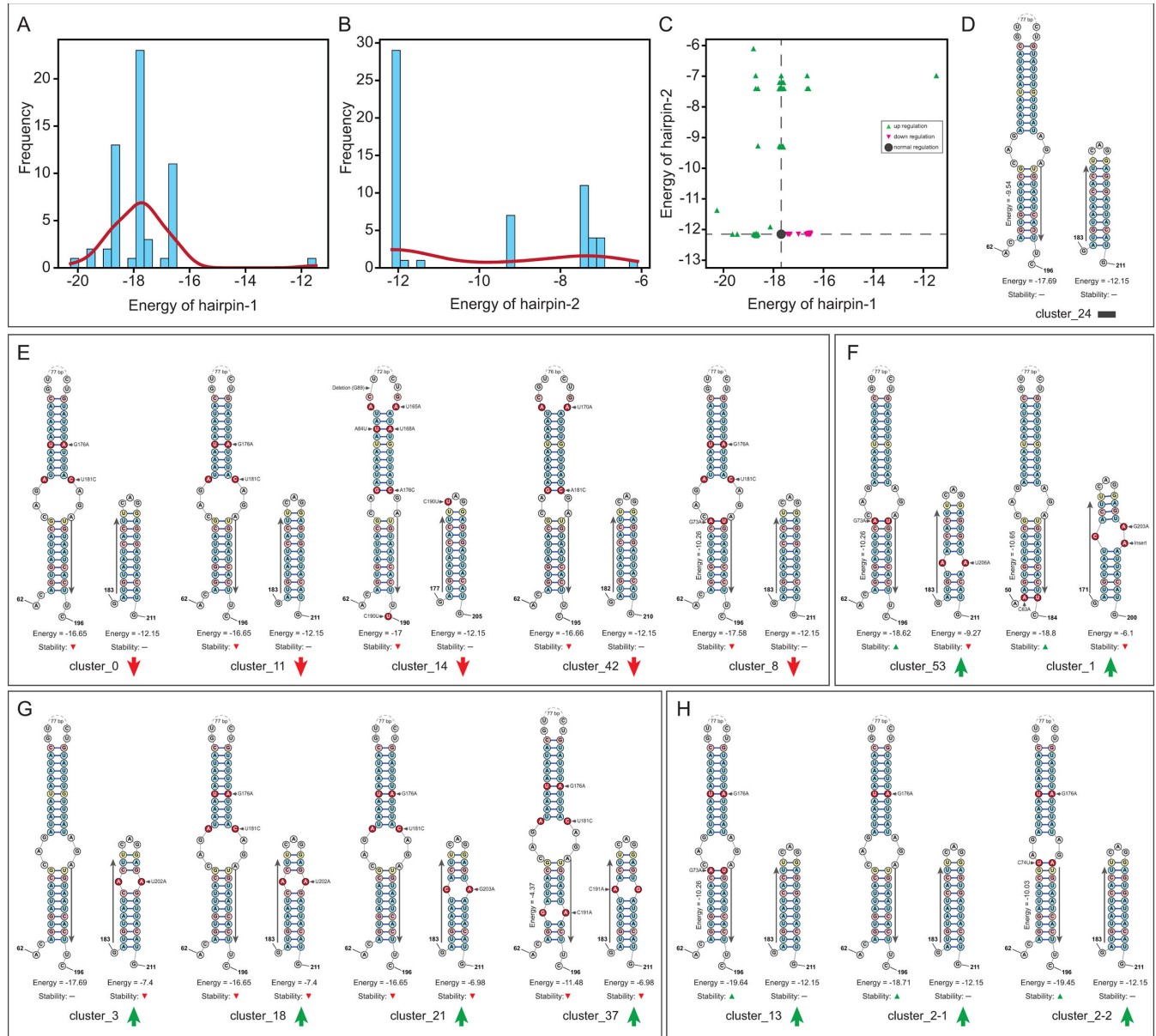

**FIG 5** Variations in the stability of alternative RNA hairpin structures in the *ermB* upstream sequence clusters. (A) Distribution of free energy for the hairpin-1 structure across all upstream sequence clusters. (B) Distribution of free energy for the hairpin-2 structure across all upstream sequence clusters. (C) A scatter plot depicting the free energy values of hairpin-1 and hairpin-2 structures across various upstream sequence clusters. (D) The two hairpin structures are formed by the reference sequence (cluster_24). (E) The two hairpin structures formed by the representative clusters with the prediction of decreased *ermB* expression inducibility. (F, G, H) The two hairpin structures formed by the representative clusters with the prediction of increased *ermB* expression inducibility. In the RNA secondary structure, G-C pair, A-U pair, and G-U pair are colored in light red, light blue, and light yellow, respectively, while the pairs with alleles (mutations) are shown in red and labeled with their positions. The free energy and structural stability of hairpin-1 (left) and −2 (right) are shown at the bottom of each hairpin structure. The sub-energy of lower-half hairpin-1 structures is shown on the right side of the hairpin-1 structure if they are different from the reference group. The prediction of *ermB* expression inducibility for each cluster is present next to each cluster name, lower inducibility is depicted as a red downward arrow, and higher inducibility is depicted as a green upward arrow.

or decreased stability of this structure, with a wide range of local free energy from −20.26 (more stable) to −11.48 (less stable) (Fig. 5A). Interestingly, for hairpin-2 (Fig. 5B), majority of clusters show a comparable level of stability as the reference cluster, and others exhibited increased free energy (from −12.15 to −6.1), indicating a trend towards instability of the hairpin-2 structures. It is worth mentioning that the reference

cluster_24 exhibited inducible and moderate *ermB* gene expression (corresponding to a weak AR or MLS-sensitive phenotype) (see below). Furthermore, when considering both factors together, we observe that these clusters feature diverse combinations of potential hairpin-1 and hairpin-2 structures. These variations can potentially result in distinct gene expression inducibility outcomes, deviating from that of the reference *ermB* variant (cluster_24) (Fig. 5C and D). The prediction rule is based on the relative free energy of hairpin structures compared to that of the reference variant (Cluster_24, Fig. 5D): weaker hairpin-1 and/or more stable hairpin-2 imply lower inducibility, while stronger hairpin-1 and/or weaker hairpin-2 suggest higher inducibility. Consequently, we categorize these combinations of hairpin-1 and hairpin-2 pairs into four groups.

Specifically, several clusters (Fig. 5E), such as cluster_0, cluster_11, cluster_14, and cluster_42, exhibited the same local free energy for hairpin-2 structure as the reference cluster_24. However, their hairpin-1 structures displayed higher local free energy values (indicating less stability) compared to the reference. This suggests that in these clusters, the formation of hairpin-2 is stronger due to reduced competition from hairpin-1, resulting in a weaker *ermB* inducibility, in the presence of antibiotics, and a likely "antibiotic sensitive" phenotype. In clusters 53 and 1 (Fig. 5F), local free energy values for hairpin-2 were significantly increased (indicating less stability), and the free energy values for hairpin-1 were slightly decreased (indicating more stability), suggesting a tendency to form hairpin-1 and a stronger *ermB* expression inducibility (leading to a resistant phenotype). Some clusters (Fig. 5G), such as cluster_3, _18, _21, _37, feature the same hairpin-1 structure, but their hairpin-2's stability decreased, still indicating a slightly stronger *ermB* inducibility and a more resistant phenotype. Finally, other clusters (Fig. 5H), such as cluster_13 and cluster_2, feature the same hairpin-2 structure but have more stable hairpin-1 structures, indicating a stronger *ermB* inducibility and a more resistant phenotype.

Thus, by specifically investigating the RNA secondary structures within the *ermB* upstream regions, we demonstrated that the combination of different alleles within the identified *ermB* upstream clusters has the potential to influence the gene expression inducibility by modulating the formation and stability of the hairpin-1 or hairpin-2 structures. This, in turn, allows us to establish a predictive link between allele combinations and the phenotype of antibiotic resistance.

## Experimental validation

To validate the accuracy of our predictions concerning the antibiotic inducibility of *ermB* expression and their effect on the AR phenotype under various allele combinations, we initially reviewed published studies reporting the sequencing of *ermB* genomic elements in bacterial isolates exhibiting MLS resistance phenotypes (51–54). In all the five instances analyzed, our predictions regarding the MLS resistance phenotype based on these genomic elements, specifically belonging to Cluster_2 (51–53) and Cluster_9 (54), were consistent with experimental observations.

To bolster the validation process, we randomly selected 11 *ermB* clusters along with their representative upstream sequences. We constructed these sequences with the same *ermB* gene. These constructs were then introduced into the neutral chromosomal site of *Staphylococcus aureus* JE2 cells, a strain sensitive to macrolide (erythromycin, ERY) and lincosamide (clindamycin, CLN) antibiotics. In our experimental setup, the wild-type *ermBL-ermB* reference construct (KES29), serving as the sensitive (weak resistance) control, allowed cells to grow in the presence of low MLS concentrations. Conversely, a construct denoted as KES30d (20), which carries a nonsense mutation at Arg-7 position and is abrogated in macrolide-inducible ribosome stalling, served as the hyperresistant control, sustaining cell growth even in the presence of high antibiotic levels.

Our observations revealed that all selected *ermB* upstream sequence (*ermBL-ermB* region) variants facilitated the inducible expression of *ermB* upon exposure to antibiotics (Fig. 6A) and did not significantly impair cell growth *in vitro* (Fig. 6B). However, these variants exhibited distinct AR phenotypes (Table 1). Specifically, two variants (Cluster_0

and Cluster_53) only supported cell growth in the presence of low antibiotic concentrations, resembling the reference construct, and thus displayed a sensitive (or weak resistance) phenotype. By contrast, the remaining nine variants allowed cell growth in the presence of high antibiotic levels, indicating an enhanced resistance phenotype. Remarkably, our experimental results closely aligned with our earlier predictions (Table 1) regarding the inducibility of *ermB* upstream regulatory regions in 10 out of 11 cases, achieving a significant 91% accuracy rate. These results strongly support the predicted role of allele combinations in governing the antibiotic inducibility of *ermB* expression and MLS phenotypes by influencing the formation and stability of upstream hairpin structures.

It is worth noting that among the constructs, specifically cluster_37, _53, and _1, although they displayed inducible expression of *ermB* in response to antibiotics, they also maintained a certain level of basal (or constitutive) expression of *ermB* even without antibiotic exposure. This suggests a less stringent regulation of *ermB* translation in these cases. Indeed, the stability of their hairpin-2 has significantly decreased. It is possible that the formation of hairpin-2 has been largely compromised when competing with the formation of hairpin-1 in these cases. This again supports the role of RNA structures in regulating the *ermB* expression.

## Emergence and evolution of *ermB*-associated antibiotic resistance in various ecological niches

Our analysis enables the classification of *ermB* loci as either MLS sensitive or resistant. We sought to understand the emergence and evolutionary trajectory of these loci, along with their associated MLS phenotypes across their ecological niches. We constructed the evolutionary relationships of all *ermB* upstream sequence clusters and the corresponding protein sequences. As depicted in Fig. 7, both the protein and nucleotide sequence phylogenies exhibit a similar overall structure, with cluster_9 forming a distinct clade, cluster_14 and cluster_42 constituting the second clade, and all others forming a larger,

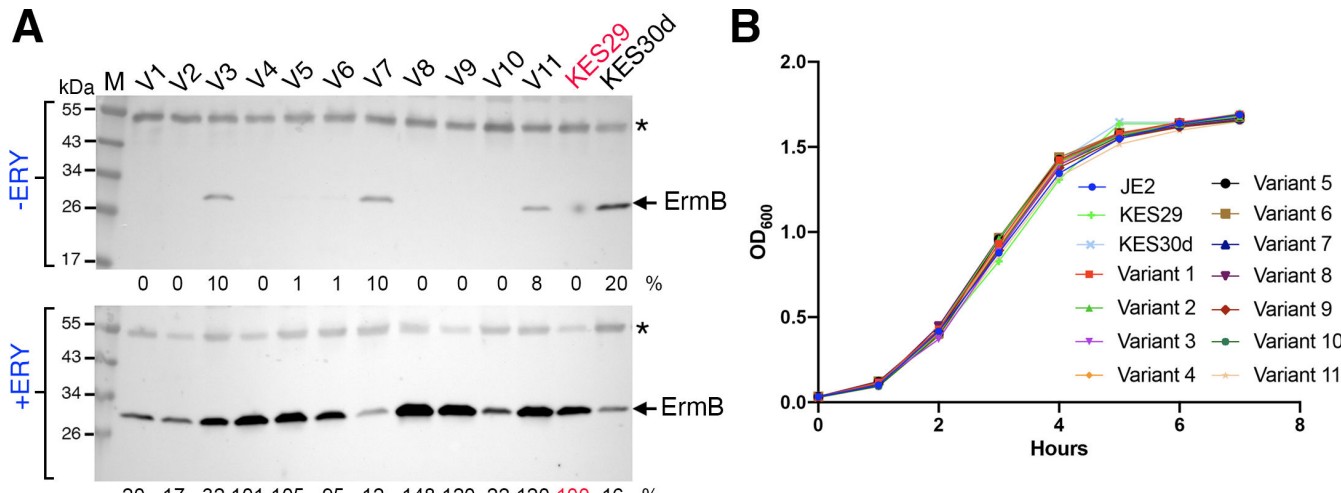

**FIG 6** Mutations in the *ermBL* and the noncoding region affect ErmB inducibility but do not significantly compromise cell growth *in vitro*. (A) Western blot analysis of ErmB abundance. *S. aureus* cells harboring different *ermBL-ermB* variants were grown at 37°C to OD$_{600}$ 0.6 in tryptic soy broth medium. The cultures were split into halves, one portion was treated with 70% ethanol (mock control) and the other portion was treated with final 1 µg/mL erythromycin (ERY). Cells were grown for an additional 90 min and harvested. A total of 0.15 Abs$_{280}$ units of cell lysates were resolved on a 4-20% SDS-PAGE, and immunoblotting was performed with anti-ErmB (1/1,000). An asterisk indicates non-specific cross-reaction and serves as a loading control. The intensities of ErmB were quantitated by ImageJ, normalized by the nonspecific band, and compared relative to the *ermBL*$^{WT}$-*ermB*$^{WT}$ (KES29) in two independent biological replicates. (B) Growth kinetics of parental *S. aureus* JE2 (*ermBL-ermB* minus) and strains carrying the ERY-inducible WT *ermBL-ermB* (KES29), constitutively expressed *ermBL*$^{R7stop}$-*ermBL* (KES30d) and variants 1–11. The optical density (OD$_{600nm}$) of tryptic soy broth (TSB) cultures was measured on a Tecan SPARK microplate reader equipped with a humidity chamber at 37°C. All experiments were repeated at least twice.

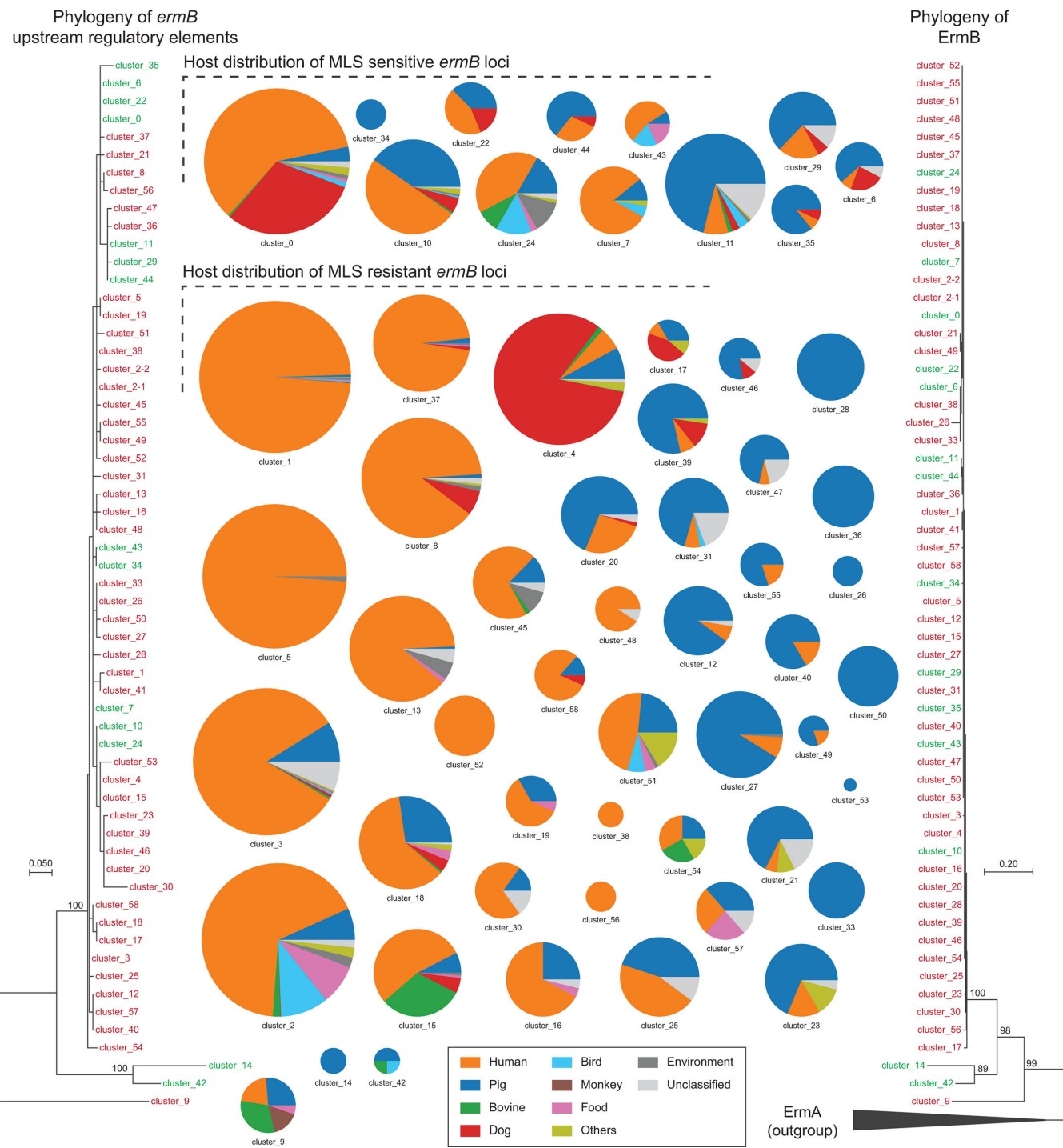

**FIG 7** Phylogeny and host distribution of either MLS-sensitive or MLS-resistant *ermB* loci. The phylogeny of the representative sequences from major clusters of *ermB* upstream region was inferred by the maximum likelihood method based on GTR model, while the phylogeny of their corresponding ErmB protein sequences was inferred using the maximum likelihood method based on the JTT model. Both trees with the highest log-likelihood are presented, in which the internal nodes with significantly supported values are shown. Each sequence is labeled with its cluster number and highlighted in either red (MLS-resistant loci) or green (MLS-sensitive loci). The size of the pie graph represents the sequence number of the corresponding cluster and is labeled with their cluster number at the bottom. The categories for the hosts are color-coded and displayed at the bottom.

third clade. Notably, MLS-sensitive and -resistant *ermB* loci are not segregated into distinct groups; rather, they are intermixed across the evolutionary tree, particularly

within clade 3. Furthermore, the short branch length within clade 3 implies a brief divergence time for this group. This observation, coupled with the mixed distribution of the MLS-sensitive and -resistant loci with the clade 3, thus suggests that MLS resistance likely emerged rapidly and independently multiple times during the evolution of *ermB*.

Furthermore, we examined the distribution of hosts from which the bacteria were isolated. The majority of the clusters are found in the bacteria isolated from a wide range of animal hosts, supporting an extensive spreading history. However, a significant number of MLS-resistant clusters are prevalent in human and pig populations, which have a long history of MLS exposure (Fig. 7). This strongly implies that the emergence and spread of MLS-resistant loci can be attributed to selection pressures resulting from widespread antibiotic use in these populations. Notably, two likely ancestral clusters, cluster_14 and cluster_42, are absent in human populations, further supporting this hypothesis. It is worth noting that while cluster_9 occupies a basal position in the tree, it may represent a recently derived version, given the limited sampling of genomes in the clade, and the ancestral MLS-sensitive locus is likely not yet sequenced.

## DISCUSSION

The increasing prevalence of antibiotic resistance is a major global public health concern, necessitating a thorough investigation into its underlying molecular mechanisms. While extensive research has focused on the relationship between AR phenotypes and specific AR genes, our understanding of how non-coding genomic allelic variations govern AR gene expression and determine AR phenotypes remains limited. One example of such non-coding regulation of AR mechanisms is observed in *erm*-associated MLS resistance, where the antibiotic-inducible expression of the *erm* gene is influenced by the upstream sequence elements containing diverse short uORFs and palindromic repeats.

Through an in-depth examination of the *erm* system, we have systematically investigated the emergence and evolution of upstream sequence features, aiming to elucidate the regulatory mechanisms. Our findings reveal that across the *erm* family, various *erm* subfamilies employ distinct upstream regions characterized by unique uORFs and palindromic regions. According to the principle of parsimony, this evidence suggests that the ancestral state of *erm* gene regulation lacked these upstream regulatory elements (both uORFs and palindromic regions), and the observed sequence features associated with *ermB, ermC,* and other subfamilies likely developed independently during evolution. This discovery aligns with our recent revelation that the *erm*-catalyzed product, $m^6A2058$ on the 23S rRNAs, negatively affects general translation and *S. aureus* fitness (20, 50). Therefore, the utilization of uORFs and palindromic regions may represent an evolutionary adaptation for bacteria to respond rapidly to antibiotic stress without compromising their functions.

The availability of population-wide genome sequence data also enables us to further examine the evolution of *erm* gene regulation once the uORF/palindromic region configuration is established. In this context, we focused on the *ermB* gene due to its relatively well-understood antibiotic-inducible mechanism. Our research uncovered those mutations in the upstream regions of *ermB* are not randomly distributed; instead, specific positions exhibit elevated variations. By performing a clustering analysis of over 20,000 sequences and subsequent RNA structure predictions, we established a connection between allele combinations and the formation and stability of hairpin structures. These hairpin structures directly influence the antibiotic inducibility of *ermB* and AR phenotypes. In addition, our data suggest a likely evolutionary process from the ancestral upstream sequence variant with low MLS resistance to multiple independent new variants with high MLS resistance within the *ermB* subfamily. Importantly, a host distinction exists between the ancestral sensitive upstream sequence variants and the newly evolved resistant variants, indicating that this transition is attributed to the selective pressure of antibiotic exposure. While the evolutionary changes in AR have long been thought to be a continuous and ongoing process, for the first time we provided the evidence at the genomic allelic level.

Furthermore, our study introduces a practical computational strategy for dissecting the relationship between the *erm* gene regulatory region and MLS-related AR prediction. Predicting AR phenotypes from genomic data is crucial in modern healthcare and infectious diseases. Current bioinformatics strategies primarily rely on the profiles of specific AR genes, forming the basis for numerous studies and bioinformatics tools (5, 8–11). However, a significant challenge arises when attempting to infer resistance phenotypes from non-coding regions of the genome, given their sequence complexity and the difficulty in deciphering their functional significance. In our study, we successfully harnessed the power of machine learning (ML) clustering techniques to facilitate the functional dissection of these non-coding regions. The population-wide genomic data in this study are characterized by a single, a few nucleotide substitutions, or indels (insertion or deletion) in the sequences, displaying high overall sequence similarity (~96% identity). Such similarity and large data sets have posed a significant challenge for traditional clustering methods, including sequence similarity-based CD-HIT (55) and phylogenetic classification (56). However, ML algorithms offer an unprecedented advantage in this regard: (1) they can identify subtle and hidden patterns within large data sets that may not be discernible through the traditional clustering methods and (2) with advanced algorithms, ML typically achieves higher classification accuracy compared to traditional methods, particularly in cases where manual classification may be challenging or ambiguous (57). Indeed, the patterns and relationships discovered by our ML method allow us to achieve a remarkable 91% accuracy rate in predicting antibiotic resistance phenotypes. Our strategy represents a substantial advancement in the ability to understand and combat antibiotic resistance. Importantly, several other antibiotic resistance genes (58), such as *cat* (involved in chloramphenicol detoxification) (59), *cmlA* (associated with multidrug efflux pumps) (60), *tlrB* (conferring resistance to tylosin antibiotics) (61), and *hflXr* (an antibiotic dissociation factor) (62), *msrD* (ketolide resistance) (63), have been found to utilize ribosome stalling mechanisms for gene regulation. These genes exhibit upstream regulatory regions containing uORFs and palindromic repeats, akin to the regulatory elements observed in *erm* genes. Thus, our ML-based approach can be seamlessly extended to these genes, offering a robust framework for studying the mechanisms and predicting AR phenotypes across a broader spectrum of non-erm resistance systems prevalent in the microbial world.

In conclusion, our study not only provides a practical solution for predicting MLS-related AR phenotypes from non-coding regions but also lays the groundwork for the development of more advanced ML methods to accurately predict the emergence of antimicrobial resistance across various bacterial species.

## ACKNOWLEDGMENTS

This research was supported by the National Institutes of Health grant R01AI150986 (to MNFY and DZ) and Department of Defense W81XWH-18-1-0122 (to MNFY).

## AUTHOR AFFILIATIONS

[1]Department of Biology, College of Arts and Sciences, Saint Louis University, St. Louis, Missouri, USA

[2]Department of Microbiology-Immunology, Northwestern University Feinberg School of Medicine, Chicago, Illinois, USA

[3]Program of Bioinformatics and Computational Biology, Saint Louis University, St. Louis, Missouri, USA

## AUTHOR ORCIDs

Yongjun Tan  http://orcid.org/0009-0006-2361-1498
Alexandre Le Scornet  http://orcid.org/0000-0002-7572-077X
Mee-Ngan Frances Yap  http://orcid.org/0000-0003-4213-4050
Dapeng Zhang  http://orcid.org/0000-0001-8535-7620

## FUNDING

| Funder | Grant(s) | Author(s) |
|---|---|---|
| HHS | National Institutes of Health (NIH) | R01AI150986 | Mee-Ngan Frances Yap |
| | | Dapeng Zhang |
| U.S. Department of Defense (DOD) | W81XWH-18-1-0122 | Mee-Ngan Frances Yap |

## AUTHOR CONTRIBUTIONS

Yongjun Tan, Conceptualization, Data curation, Formal analysis, Investigation, Methodology, Resources, Software, Validation, Visualization, Writing – original draft, Writing – review and editing | Alexandre Le Scornet, Investigation, Methodology, Resources, Validation, Visualization, Writing – review and editing | Mee-Ngan Frances Yap, Conceptualization, Funding acquisition, Methodology, Project administration, Supervision, Validation, Writing – review and editing | Dapeng Zhang, Conceptualization, Data curation, Funding acquisition, Methodology, Project administration, Resources, Supervision, Visualization, Writing – original draft, Writing – review and editing

## DATA AVAILABILITY

The retrieved genome data, alignments, specific analysis procedures, and Python3 codes are available on GitHub (https://github.com/bioconflict/ClusteringAnalysis_ermB_2024.git).

## ADDITIONAL FILES

The following material is available online.

### Supplemental Material

**Supplemental tables (mSystems00430-24-s0001.xlsx).** Tables S1 and S2.

### Open Peer Review

**PEER REVIEW HISTORY (review-history.pdf).** An accounting of the reviewer comments and feedback.

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
