## [Reviewer comments · mSystems]

Machine learning-based classification reveals distinct clusters of non-coding genomic allelic variations associated with Erm-mediated antibiotic resistance

Yongjun Tan, Alexandre Scornet, M.-N. Frances Yap, and Dapeng Zhang

Corresponding Author(s): Dapeng Zhang, Saint Louis University

Review Timeline:

Submission Date:	March 25, 2024
Editorial Decision:	April 26, 2024
Revision Received:	May 17, 2024
Accepted:	June 5, 2024

Editor: Youjun Feng

Reviewer(s): The reviewers have opted to remain anonymous.

Transaction Report:

DOI: <https://doi.org/10.1128/msystems.00430-24>

Re: mSystems00430-24 (Machine learning-based classification reveals distinct clusters of non-coding genomic allelic variations associated with Erm-mediated antibiotic resistance)

Dear Dr. Dapeng Zhang:

Revision Guidelines

Sincerely,
Youjun Feng
Editor
mSystems

Reviewer #1 (Comments for the Author):

The manuscript entitled "Machine learning-based classification reveals distinct clusters of non-coding genomic allelic variations associated with Erm-mediated antibiotic resistance" investigate regulatory mechanism of the AR gene known as RNA methyltransferase-encoding erm, which confers resistance to antibiotics belonging to the macrolides, lincosamides, and streptogramins groups (MLS phenotype). The authors found that ermB and other erm genes have independently developed

distinct configurations of uORFs and palindromic repeats during evolution. They also discovered many alleles cooperatively influenced the stability of alternative hairpin structures, thereby directly affecting ermB expression inducibility and MLS phenotypes.

Overall, I find the motivation of this study was well justified. The work is interesting.

However, the manuscript can be further improved in the following aspects.

Major issues:

1. Risk levels of ARGs should be identified and discussed.
2. To increase the transparency of the analyses, the authors should provide example source codes and the metadata of all samples. All the analysis pipeline and scripts should be deposited in GitHub.
3. When the abbreviation first appears, please use the full name meanwhile.
4. More details about the software, including specific parameters, should be provided in the Methods section to enable readers to repeat the analysis.
5. It appears that the data have not been released as of now.
6. The Abstract is too long and sentence needs to be rewritten and re-fine.

Reviewer #2 (Comments for the Author):

In face of surging antibiotic resistance, most current studies solely focused on the functionality of the coding regions of antibiotic resistance genes. In this investigation, Tan and coworkers have elucidated the evolutionary emergence of these regulatory elements of ARGs using machine learning with erm gene as proof-of-concept. The study is quite interesting and informative to shed light on the evolutionary trajectory that drives the emergence and development of MLS-induced resistance. The MS is well organized. However, there are some issues have to be concerned before consideration of publication.

1. The exact experimental details were missing in the current abstract. The authors are suggested to bring more insights of present study instead of background information.
2. Line 186, please describe the One-Hot encode method in alignment data transform in detail.
3. Please note the version of the software or program used throughout the MS.
4. As we know, ORF prediction by software sometimes bring bias like high false positive rates. Since the length of uORFs is generally short, so false positive of the prediction is expected to be high. Have the authors performed the corresponding experiment to provide evidences of the uORFs, such as ribosome profiling or proteome/ peptidome?
5. What is the rationale behind using deleted the last 12 nucleotides in ermB upstream sequences to make RNA secondary structure prediction?
6. It will be better if you use translome technology to evaluate the translation of ermB mRNA instead of protein quantification.
7. The authors are suggested to include a part of discussion for the rationale and benefit to use machine learning.
8. The English writing of MS has to be carefully polished.

Reviewer #1 (Comments for the Author):

The manuscript entitled "Machine learning-based classification reveals distinct clusters of non-coding genomic allelic variations associated with Erm-mediated antibiotic resistance" investigate regulatory mechanism of the AR gene known as RNA methyltransferase-encoding erm, which confers resistance to antibiotics belonging to the macrolides, lincosamides, and streptogramins groups (MLS phenotype). The authors found that ermB and other erm genes have independently developed distinct configurations of uORFs and palindromic repeats during evolution. They also discovered many alleles cooperatively influenced the stability of alternative hairpin structures, thereby directly affecting ermB expression inducibility and MLS phenotypes. Overall, I find the motivation of this study was well justified. The work is interesting. However, the manuscript can be further improved in the following aspects.

Response: We greatly appreciate your positive evaluation.

Major issues:

1. Risk levels of ARGs should be identified and discussed.

Response: The risk levels of ARGs, particularly the MLS resistance genes, have been addressed in the first paragraph and the last paragraph of the "Introduction" section.

2. To increase the transparency of the analyses, the authors should provide example source codes and the metadata of all samples. All the analysis pipeline and scripts should be deposited in GitHub.

Response: We have uploaded our source codes and metadata to GitHub (https://github.com/bioconflict/ClusteringAnalysis_ermB_2024.git).

3. When the abbreviation first appears, please use the full name meanwhile.

Response: We have carefully reviewed our manuscript to ensure that we include the full name before introducing an abbreviation.

4. More details about the software, including specific parameters, should be provided in the Methods section to enable readers to repeat the analysis.

Response: Thanks for the comments. We now include the detailed information regarding the program version and specific parameters, when the default parameters are not used.

5. It appears that the data have not been released as of now.

Response: Thank you for your comments. The data is now publicly available via GitHub (https://github.com/bioconflict/ClusteringAnalysis_ermB_2024.git).

6. The Abstract is too long and sentence needs to be rewritten and re-fine.

Response: Based on the feedback from both you and the other reviewer, we have significantly revised the abstract to make it more concise and informative.

Reviewer #2 (Comments for the Author):

In face of surging antibiotic resistance, most current studies solely focused on the functionality of the coding regions of antibiotic resistance genes. In this investigation, Tan and coworkers have elucidated the evolutionary emergence of these regulatory elements of ARGs using machine learning with *erm* gene as proof-of-concept. The study is quite interesting and informative to shed light on the evolutionary trajectory that drives the emergence and development of MLS-induced resistance. The MS is well organized. However, there are some issues have to be concerned before consideration of publication.

Response: Thank you for your positive evaluation.

1.The exact experimental details were missing in the current abstract. The authors are suggested to bring more insights of present study instead of background information.

Response: Based on your comments, we have significantly revised the abstract to include more experimental details.

2.Line 186, please describe the One-Hot encode method in alignment data transform in detail.

Response: Thank you for your comments. We have now included detailed information about the One-Hot encoding method under the “Methods” section: “Based on the *ermB* upstream sequence alignment, we first transformed the alignment data using the One-Hot encoding method to make it suitable for machine learning algorithms. In the alignment, nucleotide types are categorical data represented by A, T, G, C, and gap (-). One Hot encoding converts each category into a distinct numerical vector: A is encoded as (1,0,0,0); T as (0,1,0,0); C as (0,0,1,0); G as (0,0,0,1); and Gap (-) as (0,0,0,0).”

3.Please note the version of the software or program used throughout the MS.

Response: As per your suggestion, we now include the specific version information for the programs/software that were used in this study.

4. As we know, ORF prediction by software sometimes bring bias like high false positive rates. Since the length of uORFs is generally short, so false positive of the prediction is expected to be high. Have the authors performed the corresponding experiment to provide evidence of the uORFs, such as ribosome profiling or proteome/ peptidome?

Response: We agree with the reviewer that short ORF prediction is error-prone and requires additional validation. One of the senior authors had previously used ribosome profiling (Ribo-seq) to map the unannotated and misannotated ORFs in *Staphylococcus aureus* (the organism used in this study), revealing many small uORFs after experimentally confirming the production of the corresponding small proteins (PMID: 25313041, PMID: 27001516, PMID: 38252661). Applying a genome-wide Ribo-seq to validate uORF prediction in this study is unnecessary for the following reasons: this study solely focuses on the uORF of *erm* genes, the two-gene *erm* operon consisting of the uORF and its downstream *erm* has been well-studied, and the translation of *erm* uORFs and the uORF protein products have been detected by mass spectrometry (PMID: 24239289), cryo-EM structures of the uORF trapped ribosomes (PMID: 27380950, PMID: 25306253, PMID: 24961372), toeprinting (PMID: 24662426, PMID: 24961372, PMID: 21292164), cell-free translation (PMID: 18439898, PMID: 27645242), and translational fusion of various reporter genes (PMID: 26727240, PMID: 24239289, PMID: 20676057, PMID: 34262551, PMID: 27645242). Note that this reference list is not exhaustive.

5. What is the rationale behind using deleted the last 12 nucleotides in *ermB* upstream sequences to make RNA secondary structure prediction?

Response: Thank you for your comments. The most important feature of the *ermB* gene regulation is the use of the alternative RNA hairpin structures. However, the current RNA structure prediction program cannot predict such alternation of the two RNA structures; it will only provide one model that is energy optimized. In other words, when we use the full-length *ermB* upstream sequence, the program will only predict the formation of hairpin-2, not the hairpin-1, for most of the cases. Thus, to predict the formation of the hairpin 1 structure, we remove the last 12 nucleotides to prevent formation of the hairpin-2 structure. This way will allow us to compute the energy of both RNA hairpin structures, by which we can infer the regulation of *ermB* expression. The method section on this has been revised accordingly: "Specifically, we used full length *ermB* upstream sequences (1-211 bp) to predict the RNA secondary structure, resulting in all sequences only forming hairpin-2 structure but not hairpin-1. Thus, we next used the truncated version, which deleted the last 12 nucleotides in *ermB* upstream sequences (1-199 bp), to prevent the formation of the hairpin-2 and allow prediction for the hairpin-1 structure."

6. It will be better if you use translome technology to evaluate the translation of *ermB* mRNA instead of protein quantification.

Response: We agree with the reviewer that Ribo-seq can faithfully capture the global translational status. However, this transcriptome technology does not directly measure protein abundance, which could be greatly influenced by protein stability. Ribosome position and ribosome density on the mRNA, as revealed by Ribo-seq profiles, do not always show a positive correlation to the actual protein levels due to protein turnover. Likewise, allelic variations could also impact mRNA stability that is undetectable in Ribo-seq unless a parallel total mRNA-seq is performed. Our current work aims to link the abundance of ErmB methyltransferase to the allelic variation and the magnitude of MLS resistance, thus we think that measuring the levels of ErmB by Western blot (a targeted approach) is more appropriate than using the deep sequencing-based Ribo-seq. Furthermore, we focus on the expression output of only one gene operon, i.e. *ermBL-ermB*, applying a genome-wide tool such as Ribo-seq is both labor intensive and cost inefficient. We do appreciate your suggestions.

7. The authors are suggested to include a part of discussion for the rationale and benefit to use machine learning.

Response: Thank you for your suggestion. We have added a discussion section to address the benefits of using the machine learning method in the study: The population-wide genomic data in this study are characterized by single or few nucleotide mutations or indels (insertion or deletion) in the sequences, displaying high sequence similarity (~96% identity). Such highly similar but large dataset has posed challenges for traditional clustering methods, such as sequence similarity-based CD-HIT [49] and phylogenetic classification [50]. However, ML algorithms offer advantages in this regard: 1) they can identify subtle and hidden patterns within the large datasets that may not be discernible through traditional clustering methods; and 2) with advanced algorithms, ML typically achieves higher classification accuracy compared to traditional methods, particularly in cases where manual classification may be challenging or ambiguous [51].

8. The English writing of MS has to be carefully polished.

Response: Thank you for your comments. The revised manuscript has been edited by several colleagues to enhance readability.

Re: mSystems00430-24R1 (Machine learning-based classification reveals distinct clusters of non-coding genomic allelic variations associated with Erm-mediated antibiotic resistance)

Dear Dr. Dapeng Zhang:

Your manuscript has been accepted, and I am forwarding it to the ASM production staff for publication. Your paper will first be checked to make sure all elements meet the technical requirements. ASM staff will contact you if anything needs to be revised before copyediting and production can begin. Otherwise, you will be notified when your proofs are ready to be viewed.

Sincerely,
Youjun Feng
Editor
mSystems

Reviewer #1 (Comments for the Author):

The authors have addressed all my concerns. I have no further comments.

Reviewer #2 (Comments for the Author):

The authors addressed previous well, I endorse publication of MS of current version